# Molecular characterization of *Cryptosporidium* spp. from humans in Ethiopia

**Ambachew W. Hailu**[1]*, **Abraham Degarege**[2], **Haileeyesus Adamu**[3], **Damien Costa**[4], **Venceslas Villier**[4], **Abdelmounaim Mouhajir**[4], **Loic Favennec**[4], **Romy Razakandrainibe**[4], **Beyene Petros**[1]

**1** Department of Microbial Cellular and Molecular Biology, Biomedical Sciences Stream Addis Ababa University, Addis Ababa, Ethiopia, **2** Department of Epidemiology, University of Nebraska Medical Center College of Public Health, Omaha, Nebraska, United States of America, **3** Institute of Biotechnology, Addis Ababa University, Addis Ababa, Ethiopia, **4** Université de Rouen Normandie, EA7510 ESCAPE, CNR Laboratoire Expert Cryptosporidioses, CHU-Rouen, Rouen, France

* ambalake@gmail.com

**Data Availability Statement:** All relevant data are within the manuscript.

**Funding:** The author(s) received no specific funding for this work.

## Abstract

Data on the distribution and genotype of *Cryptosporidium* species is limited in Ethiopia. This study examined the presence and genetic diversity of *Cryptosporidium* species circulating in Ethiopian human population. Stool samples collected from patients who visited rural (n = 94) and urban (n = 93) health centers in Wurgissa and Hawassa district, respectively, were examined for the presence of *Cryptosporidium* spp. using microscopy, nested PCR and real-time PCR. To detect infection with PCR, analysis of *18S* ribosomal RNA was performed. Subtyping was performed by sequencing a fragment of *GP60* gene. The overall prevalence of infection was 46% (n = 86) by microscope and PCR. When 48 (out of 86) PCR positive samples were genotyped, two species were identified: *C. parvum* (n = 40) and *C. hominis* (n = 8). When 15 of the 40 *C. parvum* isolates were subtyped, zoonotic subtypes of IIaA14G1R1 (n = 1), IIaA15G2R1 (n = 1), IIaA16G1R1 (n = 2), IIaA16G3R1 (n = 2), IIaA17G1R1 (n = 1), IIaA19G1R1 (n = 1), IIaA20G1R1 (n = 3), IIaA22G1R1 (n = 1), IIaA22G2R1 (n = 1), IIdA23G1 (n = 1) and IIdA24G1 (n = 1) were identified. When 6 of the 8 *C. hominis* isolates were subtyped, subtypes IaA20 (n = 5), and IdA21(n = 1) were identified. This study suggests that *C. parvum* and *C. hominis* are causes of cryptosporidiosis in human in the Wurgissa district and Hawassa in Ethiopia. Zoonotic transmission might be the main route of transmission.

## 1. Introduction

*Cryptosporidium* species are Apicomplexan protozoan that are recognized as one of the most important diarrheal pathogens affecting people worldwide, particularly in Africa [1]. The two most common circulating *Cryptosporidium* species are *Cryptosporidium hominis* and *Cryptosporidium parvum* [2]. *C.hominis* is commonly associated with human infection while *C.parvum* is linked with infection in animals, especially young ruminants [3,4].

**Competing interests:** The authors have declared that no competing interests exist.

Water, residential surfaces, and food contaminated by *Cryptosporidium* spp. may serve as sources of infection, [5,6]. The prevailing risk factors to infection and the severity of the disease include young age, undernutrition, and impaired immunity [5]. Given the frequent and close contact between animals and human in rural areas with a possible zoonotic exposure [7–9], reports showed cryptosporidiosis is the leading cause of pediatric mortality and morbidity with anthroponotic transmission [3].

The existing routine diagnostic methods utilized for detecting *Cryptosporidium* parasites are the microscopic analysis of stool smears through staining methods such as Ziehl-Nelson [10,11]. The identification of species and subtypes of *Cryptosporidium* is dependent on molecular techniques [12]. In many parts of Africa, the infrastructure for molecular characterization is not yet evolved [3] and consequently studies on the distribution of *Cryptosporidium* species, genotypes, and transmission routes are scanty in the region.

Ethiopian's population is growing rapidly (approximately 3% annually). The mixed crop-livestock system of Ethiopia carries more than 70% of the cattle population, which may increase anthroponotic and zoonotic transmission of *Cryptosporidium* [13]. In addition, dairy operations in densely populated urban and peri-urban settings, poor hygienic and sanitation conditions could create hot spots for zoonotic transmission. Indeed, a study reported *Cryptosporidium parvum* among 35 (87.5%) pre-weaned calves specimens examined in central Ethiopia [14]. However, due to the paucity of routine screening for *Cryptosporidium* spp., and absence of systematic investigation of cases by the health system, the prevalence of *Cryptosporidium* infection and genetic diversity of the parasite in human population in Ethiopia remain uncertain. To our knowledge, there is only one study that reported genetic diversity of *Cryptosporidium* spp., (*C.hominis* and *C.parvum*) among HIV/AIDS patients in Ethiopia [15]. The prevalence and genetic characterization of *Cryptosporidium* spp. in the general population remain uncertain. The objective of this study was to determine the prevalence of *Cryptosporidium* infection and identify genetic diversity of the parasite circulating in human population living in Wurgissa district and Hawassa in Ethiopia.

## 2. Materials and methods

### 2.1. Study areas

The study was conducted in health centers located in Wurgissa and Hawassa districts from January to September 2018. Wurgessa district is located in the rural area of the Amhara region in northeast Ethiopia. Hawassa district is an urban city located in the Southern Nations, Nationalities, and Peoples' Region.

### 2.2. Specimen collection and transport

Single fecal samples were collected from volunteer patients who visited Wurgessa Health Center (WHC) and Hawassa Health Center (HHC). Fresh stool samples were collected from 94 individuals at WHC and from 93 at HHC. The inclusion criterion for patients in this study was contact with domestic or wild animals. A questionnaire was administered to collect data on other potential risk factors for *Cryptosporidium* infection, such as diarrhea in other members of the household, HIV serostatus, presence of animals within the house, defecation sites, education, and drinking water supply sources. Approximately 2g of fecal sample was transported to the biomedical science laboratory of Addis Ababa University (AAU) for microscopy analysis. About 1 g of fecal material was placed in an 8 mL aliquot of 2.5% (*w/v*) potassium dichromate, thoroughly mixed, and transported to the Centre National de Reference Cryptosporidiosis (French National Reference Centre for Cryptosporidiosis) at Charles Nicole

University Hospital, Rouen (France) for characterization. Samples were kept at 4°C before DNA isolation.

## 2.3. Microscopic detection of Cryptosporidium spp. oocysts in fecal samples

After removal of the preservative through washing, the specimens were concentrated via for-malin–ethyl acetate sedimentation [16], and a thin fecal smear was examined for each speci-men after staining with modified Ziehl–Neelsen technique [17]. Briefly, slides were stained with carbol fuchsin and differentiated in 1% hydrochloric acid–alcohol (70%) for 1 min before counterstaining with 1% methylene blue for 1 min. The stained slides were examined using an oil immersion lens at 100× magnification, where oocysts stained pink to red or deep purple against a blue background. The presence or absence of *Cryptosporidium* was recorded for each stool sample examined.

## 2.4. DNA extraction, molecular detection, and subtyping

Nucleic acid was extracted from all fecal specimens using the *QIAamp Power fecal DNA kit* (Qia-gen, France) following the manufacturer's protocol. To enable the rapid detection and identifica-tion of *C. hominis* and *C. parvum*, two major species that are associated with human cryptosporidiosis, samples were screened using *18S* ribosomal RNA *(rRNA)* nested PCR and real-time PCR as described elsewhere [18]. Briefly, PCR was carried out in duplicate and consisted of two duplex reactions: a genus-specific PCR amplifying 300 bp of the *Cryptosporidium 18S rRNA* gene, duplexed (i) with a *C. parvum*-specific PCR amplifying 166 bp of the *LIB13* locus, and (ii) with a *C. hominis*-specific PCR amplifying 169 bp of the *LIB13* locus. Thermocycling conditions were as follows: 95°C for 10 min, followed by 55 cycles of 95°C for 15 s and 60°C for 60 s. Data were collected from each probe channel during each 60°C annealing/extension phase.

To correctly identify other species infecting human and to confirm results from the real-time PCR, genomic DNA extracts were subjected to a nested PCR-based sequencing protocol, targeting the *18S* ribosomal RNA *(rRNA)* gene, as described elsewhere [19]. For the primary PCR, the cycling protocol was as follows: 94°C for 5 min; followed by 40 cycles of 94°C for 30 s, 58°C for 45 s, and 72°C for 1 min; with a final extension of 72°C for 5 min. For the secondary PCR, the protocol was as follows: 94°C for 5 min; followed by 40 cycles of 94°C for 30 s, 58°C for 45 s, and 72°C for 45 s; with a final extension of 72°C for 5 min. Products were visualized in 2% agarose gels using ethidium bromide staining. Positive samples were further subtyped by DNA sequencing of the *GP60* gene.

Subtyping was performed by sequencing a fragment of the *GP60* gene. Each sample was amplified at least three times by nested PCR. Primers *AL3531* and *AL3533* were used in pri-mary PCR, and primers *AL3532* and *LX0029* were used in secondary PCR [20]. Reaction mix-tures were prepared using 5 *μL* 10× DreamTaq Buffer, 0.2 *mM* of each deoxynucleoside triphosphate, 100 *nM* of each primer, 2.5 U DreamTaq polymerase, and 5 *μL* DNA template. Additionally, 1.25 *μL* of dimethyl sulfoxide was added to the mixture. Cycle conditions were as follows: one cycle of 94°C for 3 min; 39 cycles of a denaturation step at 94°C for 45 s, an annealing step at 54°C (for both the first and the second rounds) for 45 s, and an extension step at 72°C for 1 min; with a final extension for 10 min at 72°C.

Each amplification run included a negative control (PCR water) and two positive controls (genomic DNA from *C. parvum* oocysts purchased from INRAE Centre Val de Loire-Nouzilly France, and *C. hominis* genomic DNA from a fecal specimen collected at Rouen University Hospital). Products were visualized in 2% agarose gels using ethidium bromide staining, and sequencing was used for identification and subtype confirmation. PCR amplicons were

purified using exonuclease I/shrimp alkaline phosphatase (*Exo-SAP-IT*) (USB Corporation, Cleveland, Ohio, USA). They were sequenced in both directions using the same PCR primers at 3.2 *uM* in 10 *μL* reactions with Big Dye™ chemistry in an ABI 3500 sequence analyzer (Applied 229 Biosystems, California, USA). Sequence chromatograms of each strand were examined with 4Peaks software and compared with published sequences in the GenBank data-base using the Basic Local Alignment Search Tool (BLAST; *www.ncbi.nlm.nih.gov/BLAST*).

### 2.5. Consent and ethical approval

This study was approved by the ethical clearance committee of the College of Science at Addis Ababa University. All participants were briefed about the aims of the study protocol and verbal consent obtained prior to sampling. As the procedure for obtaining stool sample from the study participants had minimal effect, the IRB approved verbal consent. Assent of the children and consent of their parent or guardian was sought.

### 2.6. Statistical analysis

*SPSS* Statistics (version 26) was used for the analysis. Prevalence of infection was compared across sociodemographic groups using chi-square or fisher exact test (when the count for at least one cell was less than 5). The combined results based on microscope, nested PCR and real time PCR was used as a 'gold standard' (True result) to calculate the sensitivity, specificity, and predictive values of the three tests in detecting *Cryptosporidium* spp. infection. Multiple logis-tic regression analysis was used to identify factors associated with *Cryptosporidium* spp. infec-tions. Kappa value was used to examine the agreement between the tests in detecting the presence of *Cryptosporidium* spp. infections. A kappa value greater than 0.81 was considered perfect agreement, and kappa value that fall between 0.61 and 0.80 were considered substantial agreement, while a value ranges between 0.41 and 0.60 was a moderate kappa agreement [21]. *P*-values less than 0.05 and 95% confidence intervals were considered statistically significant associations between sociodemographic factors and infection.

## 3. Results

### 3.1. Sociodemographic description of study participants

Of the 187 study participants, 108 (57.8%) were male, 94 (50.3%) lived in rural areas and 123 (65.8%) were illiterate. The mean age of the study participants was 31.7 years (range: 6–66 years). A total of 25 (13.4%) were children or adolescents and 162 (86.6%) were adults. Of the adults, 55.1% reported gastrointestinal symptoms with diarrhea perior to visiting the health center. From the children, 80% had a recent history of clinical signs related to the gastrointestinal tract.

### 3.2. Prevalence of Cryptosporidium infection

Based on the combined results applying microscope, nested PCR and real-time PCR, the prev-alence *Cryptosporidium* spp. infections among the study participants was 46.0%. *Cryptosporid-ium* was more prevalent in patients with no formal education and those living with HIV (Table 1). The prevalence of infection was comparable between males and females. The differ-ence in the prevalence of infection was also not significant across different age groups.

### 3.3. Performance of the nested PCR, real time PCR and microscope in detecting Cryptosporidium spp. infections

The prevalence of *Cryptosporidium* spp. infections was 17.1% (32/187), 24.6% (46/187) and 41.7% (78/187) using microscopy, nested PCR and real-time PCR, respectively. There was a

**Table 1. Prevalence of *Cryptosporidium* infection.**

| Chracterstics | Categories | Number examined | Microscope | Real time PCR | Nested PCR | Combined micrscope, nested PCR and real time PCR |
|---|---|---|---|---|---|---|
| **Age** | 0–9 | 4 | 25.0% | 25.0% | - | 25.0% |
| | 10–19 | 21 | 9.5% | 52.4% | 42.9% | 57.1% |
| | 20–30 | 78 | 15.4% | 35.9% | 16.7% | 41.0% |
| | 31–40 | 43 | 11.6% | 39.5% | 18.6% | 39.5% |
| | 41–50 | 24 | 29.2% | 50.0% | 37.5% | 58.3% |
| | ≥51 | 17 | 29.4% | 47.1% | 41.2% | 58.8% |
| **p-value** | | | 0.27 | 0.53 | 0.02 | 0.32 |
| **Gender** | Female | 79 | 15.2% | 40.5% | 20.3% | 45.6% |
| | Male | 108 | 18.5% | 42.6% | 27.8% | 46.3% |
| **p-value** | | | 0.70 | 0.88 | 0.30 | 1.00 |
| **Education level** | No formal Education | 123 | 18.7% | 51.2% | 29.26% | 56.10% |
| | Formal Education | 64 | 14.06% | 23.43% | 15.62% | 26.56% |
| **p-value** | | | 0.42 | 0.04 | 0.001 | 0.001 |
| **Location** | Wurgissa | 94 | 16.0% | 40.4% | 20.2% | 45.7% |
| | Hawasa | 93 | 18.3% | 43.0% | 29.0% | 46.2% |
| **p-value** | | | 0.70 | 0.76 | 0.18 | 0.99 |
| **HIV sero status** | Positive | 41 | 39.0% | 80.5% | 53.7% | 87.8% |
| | Negative | 95 | 5.3% | 20.0% | 6.3% | 24.2% |
| | Unknown | 51 | 21.6% | 51.0% | 35.3% | 52.9% |
| **p-value** | | | <0.001 | <0.001 | <0.001 | <0.001 |

statistically significant difference in the prevalence of *Cryptosporidium* spp. infection detected using microscopic, nested PCR and real-time PCR (p<0.01).

All samples detected positive for *Cryptosporidium* spp. infections by microscopy were also positive with the nested PCR and real-time PCR. However, 25 individuals detected as positive by the nested PCR and 46 samples detected positive by real time PCR were negative by microscopy. A total of 40 samples determined positive by the real time PCR were negative by the nested PCR and 8 samples detected positive by nested PCR were negative by real time PCR.

Using the combined results based on the three methods as a 'true result', sensitivity of the microscopy, nested PCR and real-time PCR in detecting *Cryptosporidium* infection was 38.6%, 51.8% and 94.0% (Table 2). The corresponding negative predictive values for these tests were 67.1%, 71.6%, and 95.4%. The specificity and positive predictive values for nested PCR were 97.1% and 93.5%, respectively. However, the specificity and positive predictive values were 100% for microscope and real time PCR. The agreement between the microscope and the combined results using the three tests to detect infection was moderate (k = 0.41). The agreement between Nested PCR and the combined results using the three tests in detecting infection was also moderate (0.51) agreement. The agreement between real time PCR and the combined results was almost perfect (k = 0.95).

**Table 2. Comparison of the performance of the nested PCR, real time PCR and microscope in detecting *Cryptosporidium* spp. infections.**

| Diagnostic methods | Prevalence of infection | Sensitivity | Specificity | Negative predictive value | Positive predictive value | Accuracy | Kappa |
|---|---|---|---|---|---|---|---|
| **Microscope** | 17.1% | 38.6% | *100%* | 67.1% | 100% | 72.7 | 0.41 |
| **Nested PCR** | 24.6% | 51.8% | 97.1% | 71.6% | 93.5% | 77.0 | 0.51 |
| **Real time PCR** | 41.7 | 94.0% | 100% | 95.4% | 100 | 97.3 | 0.95 |

### 3.4. Cryptosporidium spp. and subtypes

Genotype data for *Cryptosporidium* spp. were obtained in 48 of 86 positive PCR samples. Among those genotyped, *C. parvum* (n = 40) was frequently detected. C. *hominis* was detected in 8 samples. Subtype analysis was successfully carried out for 15 of 40 infections with *C. parvum* and 6 of 8 infections with *C. hominis*. Infections with *C. parvum* belonged to zoonotic subtype families IIa and IId. When 15 of the 40 *C. parvum* isolates were subtyped, zoonotic subtypes of IIaA14G1R1 (n = 1), IIaA15G2R1 (n = 1), IIaA16G1R1 (n = 2), IIaA16G3R1 (n = 2), IIaA17G1R1 (n = 1), IIaA19G1R1 (n = 1), IIaA20G1R1 (n = 3), IIaA22G1R1 (n = 1), IIaA22G2R1 (n = 1), IIdA23G1 (n = 1), and IIdA24G1 (n = 1) were identified. Two subtype families were identified within C. *hominis* (Ia and Id). When 6 of the 8 C. *hominis* isolates were subtyped, subtypes IaA20 (n = 5), and IdA21 (n = 1) were identified. Representative sequences were deposited in the NCBI database under accession numbers MW037825–MW037836.

### 3.5 Risk factors for Cryptosporidiosis

The occurrence of diarrhea in other members of their households (adjusted odds ratio (AOR) = 34.17, p<0.01) and the household size (AOR = 21.17, p<0.01) were positive factors for *Cryptosporidium* infection (Table 3). A total of 131 (70.5%) patients had close contact with cattle, which were mainly cows and calves in both urban and rural households. In this context, the presence of animals was a positive predictor of *Cryptosporidium* infection (AOR = 12.13, p <0.01). many of the participants also had multifaceted contact with a non-human primate and this contact was a positive predictor of *Cryptosporidium* infection (AOR = 36.26, p<0.01). In addition, Urban recreational location (AOR = 4.53, p <0.05) and HIV seropositivity (AOR = 168.22, p < 0.01) were significant factors in *Cryptosporidium* infection.

## 4. Discussion

In the present survey, the PCR based prevalence of cryptosporidiosis was 46% (86/187), which is comparable to earlier findings among HIV/AIDS patients in the northern part of Ethiopia (43.6%) [22]. However, this prevalence is considerably higher than that reported among patients with gastrointestinal symptoms (1.1%) [23], those living with HIV in southern Ethiopia (13.2%) [24], or schoolchildren in northwest Ethiopia (4.6%) [25]. The lower prevalence of cryptosporidiosis reported in the aforementioned studies in Ethiopia could be due to the less sensitivity of the microscopy procedures used for diagnosing *Cryptospordium* infection [26]. On the other hand, the high prevalence infection observed in the present study could be due to increased animal contact, overcroweded living conditions, household diarrhea, open defecation and lack access to clean water which are significant risk factor for cryptosporidiosis infection [27]. Such condition might lead to a repeated exposure of the population to Cryptosporidium oocysts and the development of an immunity state and less symptoms of the infection [28]. In other study asymptomatic oocyst shedding has been noted in apparently healthy individuals [29] which can explain a passive transfer of oocysts in human digestive system. However, the perevalence of *Cryptosporidium* determined using a microscope was significantly lower than the estimate based on PCR. A microscope may miss oocysts when the intensity of infection is low [30]. In addition, through microscope, the oocysts my appear colorless, smooth, and spherical bodies increasing the chance of missing the infection [31].

Of the isolates of infected samples that were genotyped, *C. parvum* (n = 40) and *C. hominis* (n = 8) were the only detected species. *C. parvum* isolates demonstrated 15 subtypes belonging to two subtype families (IIa and IId) and *C. hominis* showed six subtypes that belong to two subtype families (Ia and Id). The most common detected *C. parvum* subtype was IIaA20G1R1. Reports of subtype IIaA20G1R1 in humans are rare. However, the IIaA20G1R1 subtype was

**Table 3. Factors associated with *Cryptosporidium* infection characteristics.**

| Attribute | Categories | Unadjusted OR [95% CI] | Adjusted OR [95% CI] |
|---|---|---|---|
| **Age** | 0–9 | - | - |
| | 10–19 | 4.00 [0.35–45.10] | 9.45 [0.002–32960] |
| | 20–30 | 2.08 [0.20–20.97] | 6.44 [0.002–19199] |
| | 31–40 | 1.96 [0.18–20.45] | 5.96 [0.017–20261] |
| | 41–50 | 4.19 [0.38–46.50] | 1.99 [0.0005–7293] |
| | > = 51 | 4.28 [0.37–50.19], | 29.52 [0.006–132218] |
| **Gender** | Male | - | - |
| | Female | 0.97 [0.54–1.73] | 0.75 [0.22–2.59] |
| **Location** | Wurgissa | - | - |
| | Hawassa | 1.02 [0.57–1.81] | 4.53 [1.00–20.52] |
| **Education level** | No Formal education | - | - |
| | Formal education | 0.28 [0.14–0.55] | 0.47 [0.13–1.60] |
| **Family size** | ≤4 | - | - |
| | >4 | 8.48 [3.32–21.64] | 21.17 [2.89–155.12] |
| **Contact with diarrhea patient** | No | - | |
| | Yes | 8.07 [3.74–17.35] | 34.17 [7.07–165.1] |
| **Contact with apes** | No | - | - |
| | Yes | 9.44 [4.27–20.88] | 36.26 [6.02–218.3] |
| **Source of drinking Water** | Tap water | - | - |
| | Open well water | 1.57 [0.58–4.20] | 0.43 [0.03–.4.97] |
| | Stream water | 1.26 [0.50–3.20] | 0.45 [0.026–7.78] |
| **Defecation habit** | Toilet facility | - | - |
| | Open field | 0.80 [0.30–2.19] | 3.36 [0.21–53.6] |
| | Near To the river | 2.10 [0.86–5.13] | 8.13 [0.59–1104] |
| **Handwashing habit** | No | - | - |
| | Yes | 1.10 [0.62–1.96] | 0.70 [0.22–2.24] |
| **Presence of animals at home** | No | | |
| | Yes | 5.47 [2.60–11.5] | 12.13 [2.34–62.93] |
| **Presence of diarrhea** | No | - | - |
| | Yes | 2.35 [1.29–4.26] | 1.92 [0.57–6.40] |
| **HIV Serostatus** | Negative | - | - |
| | Positive | 22.54 [7.91–64.19] | 168.22 [16.19–1747] |
| | Unknown | 3.52 [1.70–7.25] | 4.53 [2.89–155.12] |

seen in water buffalo in Brazil [32] and in cattle in Serbia and Montenegro [33], Sweden [34], and Brazil [35]. The second most common subtpes of *C. parvum* identified in this study were IIaA16G1R1 and IIaA16G3R1. The IIaA16G1R1 subtype was reported in lamb, calves, and humans, as well as water sources in Romania [36], Estonia [37], and Slovakia [38]. The IIaA16G3R1 subtype was also seen in calves and goats in Spain, England and wild ponies on the Iberian Peninsula [39–41]. A study also reported a high prevalence of *C. parvum* subtypes that belong to the IIa and IId families in sheep and claves from Italy [42,43]. IId subtypes have also been identified in human samples from Egypt, Ethiopia, and Malaysia [15,44–46] and a range of animal hosts from China, such as horses and donkeys, rodents, golden takins, yaks, sheep, and goats [47,48]. Altogether, these findings may suggest human and animals as reservoirs for the *C. parvum* [49,50]. Thus, an integrated, transdisciplinary and multilevel one health approach strategies/intervention that target humans, animals and their shared environment/transmission routes would be necessarily to effectively control cryptosporidiosis in

regions endemic for *C. parvum* infection [51]. An integerated research involving veterinary, public health and environmental fields would help better understand the burden, risk factors, transmission routes of the zoonotic *C. parvum* infection and plan collaborative one health approach to treat or prevent infection in animals and humans, reduce environmental contamaination and block transmission in endemic regions [51].

In this study, two subtypes of *C. hominis* were recorded: IaA20 (5/6) and IdA21 (1/6). The latter were previously recorded in travelers in the United Kingdom (UK) returning from Africa [52,53], with no prior identification in African studies. This may suggest that C. hominis IaA20 is the most widespread subtype for the study area. There could be potential for zoonotic or anthroponotic transmission in the region. Further molecular studies from different hosts will be crucial to better understanding the epidemiology of cryptosporidiosis in Ethiopia.

In the present study, HIV infection, contact with animals, contact with non-human primates, household size (>4), and contact with diarrheal person were significantly associated with *Cryptosporidium* infection. The close proximity between human and non-human primates was found to be a positive predictor for *Cryptosporidium* infection. Though there is a dearth of epidemiological information on the association between humans and non-human primates (NHPs), studies in Uganda showed higher prevalence of *Cryptosporidium spp*. in human adapted NHPs [54] and similar subtypes recorded both in the community and NHPs [55]. In fact, wild animals (such as NHPs) are a potential source of infection as they can spread parasites to humans via direct contact or through contamination of drinking and recreational water, farms, and edible fruits and vegetables [56].

This study provides data on the genetic characterization of *Cryptosporidium* spp. in the general human population in rural and urban regions of Ethiopia where there is limited data. To our knowledge there is only one study that reported data on the genetic diversity of *Cryptosporidium* spp in human samples, focusing on HIV patients in Ethiopia [15]. However, the sample size for this study may not have enough power to test correlation of different sociodemographic factors and health condition with the risk of getting infection with *Cryptosporidium*. In addition, as genotyping was performed for small human samples, confirmation of zoonotic transmission was limited in this study.

## 5. Conclusions

This study suggests that *C. parvum* and *C. hominis* are causes of cryptosporidiosis in humans in the Wurgessa district and Hawassa in Ethiopia. The identification of the subtype IId family with high zoonotic potential in various hosts suggests that zoonotic transmission might be the main route of transmission in the study area. Studies with larger sample sizes, including animals, would be important to verify the current finding and understand the *Cryptosporidium* subtypes and possible routes of transmission in Ethiopia.

## Acknowledgments

We thank all participants in this study, without whom this work could not have existed.

## Author Contributions

**Conceptualization:** Ambachew W. Hailu, Haileeyesus Adamu.

**Data curation:** Ambachew W. Hailu, Abraham Degarege, Haileeyesus Adamu, Damien Costa, Abdelmounaim Mouhajir, Romy Razakandrainibe, Beyene Petros.

**Formal analysis:** Ambachew W. Hailu, Abraham Degarege, Damien Costa, Romy Razakandrainibe.

**Funding acquisition:** Loic Favennec, Beyene Petros.

**Investigation:** Ambachew W. Hailu, Haileeyesus Adamu, Damien Costa, Venceslas Villier, Romy Razakandrainibe.

**Methodology:** Ambachew W. Hailu, Haileeyesus Adamu, Damien Costa, Venceslas Villier, Romy Razakandrainibe.

**Project administration:** Ambachew W. Hailu, Loic Favennec, Romy Razakandrainibe, Beyene Petros.

**Resources:** Ambachew W. Hailu, Damien Costa, Loic Favennec.

**Software:** Ambachew W. Hailu, Damien Costa, Venceslas Villier, Abdelmounaim Mouhajir, Loic Favennec, Romy Razakandrainibe.

**Supervision:** Ambachew W. Hailu, Abraham Degarege, Haileeyesus Adamu, Loic Favennec, Romy Razakandrainibe, Beyene Petros.

**Validation:** Haileeyesus Adamu, Damien Costa, Abdelmounaim Mouhajir, Loic Favennec, Romy Razakandrainibe, Beyene Petros.

**Visualization:** Ambachew W. Hailu, Abraham Degarege, Damien Costa, Venceslas Villier, Abdelmounaim Mouhajir, Loic Favennec, Romy Razakandrainibe, Beyene Petros.

**Writing – original draft:** Ambachew W. Hailu, Abraham Degarege.

**Writing – review & editing:** Ambachew W. Hailu, Abraham Degarege, Romy Razakandrainibe, Beyene Petros.

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
