## [Decision Letter · Decision Letter 0]

8 Apr 2021

PONE-D-21-06475

Characterization of Cryptosporidium spp. in humans in Ethiopia

PLOS ONE

Dear Dr. Woreta,

Thank you for submitting your manuscript to PLOS ONE. After careful consideration, we feel that it has merit but does not fully meet PLOS ONE’s publication criteria as it currently stands. Therefore, we invite you to submit a revised version of the manuscript that addresses the points raised during the review process.

We look forward to receiving your revised manuscript.

Kind regards,

Maria Stefania Latrofa

Academic Editor

PLOS ONE

Journal Requirements:

2. We note you have included a table to which you do not refer in the text of your manuscript. Please ensure that you refer to Table 3 in your text; if accepted, production will need this reference to link the reader to the Table.

3. We note that Figure 1 in your submission contain map images which may be copyrighted. All PLOS content is published under the Creative Commons Attribution License (CC BY 4.0), which means that the manuscript, images, and Supporting Information files will be freely available online, and any third party is permitted to access, download, copy, distribute, and use these materials in any way, even commercially, with proper attribution. For these reasons, we cannot publish previously copyrighted maps or satellite images created using proprietary data, such as Google software (Google Maps, Street View, and Earth). For more information, see our copyright guidelines: http://journals.plos.org/plosone/s/licenses-and-copyright.

Reviewers' comments:

Reviewer's Responses to Questions

**Comments to the Author**

1. Is the manuscript technically sound, and do the data support the conclusions?

Reviewer #1: Yes

Reviewer #2: Partly

2. Has the statistical analysis been performed appropriately and rigorously? 

Reviewer #1: Yes

Reviewer #2: Yes

3. Have the authors made all data underlying the findings in their manuscript fully available?

Reviewer #1: Yes

Reviewer #2: Yes

4. Is the manuscript presented in an intelligible fashion and written in standard English?

Reviewer #1: Yes

Reviewer #2: Yes

5. Review Comments to the Author

Reviewer #1: The manuscript focus on human criptosporidiosis in Ethiopia by comparing different standard and molecular techniques. Results are impressive and deserves of being published, given their importance.

My only suggestions is to improve comparision with other realities and also in one health approach, considering also animal reservoir. These recent papers can be included into the discussion.

Line 258 and after:

Dessì, G., Tamponi, C., Varcasia, A., Sanna, G., Pipia, A.P., Carta, S., Salis, F., Díaz, P., Scala, A.

Cryptosporidium infections in sheep farms from Italy

(2020) Parasitology Research, 119 (12), pp. 4211-4218.

Díaz, P., Varcasia, A., Pipia, A.P., Tamponi, C., Sanna, G., Prieto, A., Ruiu, A., Spissu, P., Díez-Baños, P., Morrondo, P., Scala, A.

Molecular characterisation and risk factor analysis of Cryptosporidium spp. in calves from Italy

(2018) Parasitology Research, 117 (10), pp. 3081-3090.

Reviewer #2: Study provides data on the prevalence and genetic diversity of Cryptosporidium in human patients in Ethiopia. Study surely fills a gap I knowledge on Cryptosporidium species involved in human cryptosporidioses in the area, however, relatively small sample set does not allow for robust interpretation. Considering this fact, some parts of the study seem a bit over-interpreted, mainly when discussing zoonotic vs. anthroponotic transmission. It is questionable, how much the fact, that some of the genotypes are shared among animals and humans, suggests an infection from animal reservoir. For such an assumptions, results from animals living in proximity of tested subjects can be of great value.

Relatively high number of samples proved positive only in NAATs, which opens a question of passive transfer of the oocysts through human digestive system. For humans living in highly contaminated environment this can be expected. This part might deserve some explanation or justification… otherwise the “true” prevalence mentioned might be the “distorted one” (Line 198).

Paragraph in lines 308-315 raises several doubts about part of the results and mainly about conclusiveness of the study. In my opinion, its placement before Conclusions without explanations is misleading. This deserves some attention and explanations… possibly in respective parts of the results and discussions.

I can only suggest the MS for publication after some revisions that consider also interpretation of results. Some minor comments follows:

Line 55: consider replacing of “for identifying” by “for detection”

Line 66-67: mentioning fairly distant and non-zoonotic abomasal C. andersoni without any explanation is confusing.

Line 71: replace “genetic distribution” by more proper term

Line 75: the goal “identifying genetic diversity… … in Ethiopia” is over-ambitious, considering the small number of samples from a single geographic region…

Line 99 (and some other places): double check using the upper case letters

Line 175: “55.1% reported gastrointestinal symptoms with diarrhea”… not clear to me, if this was in time of sample collection or in “any” time before visiting the hospital

Line 179:”infection” not Italic

Line 241-245: it is always high to compare data from various studies with various methodology and robustness of diagnostics, however, the option, that lower prevalences can be attributable to lower performance of diagnostic procedures deserves mentioning, before searching for other explanations.

Line 249-252: not clear to me, why zoonotic transmission is considered more plausible than human-to-human

Line 252-253 requires references

Line 270-272: you say “Moreover, this subtype was recorded in immunocompetent adults in Slovakia [42] and children in Mexico [43], which provides evidence for its role in animal-to-human transmission”… this is not clear… how this relates to animal-to-human transmission? Please, clarify…

Lines 272-274: the statement about sylvatic transmission is purely speculative in given context, mentioning feral horses but also (probably?) domestic calves and goats.

Lines 283-302: this deserves some shortening, considering the sample set size and low significance of the results. In discussed aspects the study revealed nothing new.

Last but not least, please, pay attention to formatting of references

6. PLOS authors have the option to publish the peer review history of their article (what does this mean?). If published, this will include your full peer review and any attached files.

Reviewer #1: No

Reviewer #2: **Yes: **David Modrý

---

## [Author Response · Author response to Decision Letter 0]

10 May 2021

PONE-D-21-06475

Molecular characterization of Cryptosporidium spp. from humans in Ethiopia

PLOS ONE 

Dear Dr. Maria Stefania Latrofa 

Thank you for your comments and sending us the reviewers' comments. We thank also the reviewers for their constructive comments. We have revised the manuscript following the reviewers' suggestions. describe these changes in the below paragraphs. We hope that you will find our responses acceptable and we look forward to your decision.

Editors 

Response: We have checked the author guidelines for formatting of the manuscript. The revised MS meets the Plos one requirements. 

2. We note you have included a table to which you do not refer in the text of your manuscript. Please ensure that you refer to Table 3 in your text; if accepted, production will need this reference to link the reader to the Table.

Response: We thank the editor for picking this. We have insereted table 3 in the text in line # 221 

3. We note that Figure 1 in your submission contain map images which may be copyrighted. All PLOS content is published under the Creative Commons Attribution License (CC BY 4.0), which means that the manuscript, images, and Supporting Information files will be freely available online, and any third party is permitted to access, download, copy, distribute, and use these materials in any way, even commercially, with proper attribution. For these reasons, we cannot publish previously copyrighted maps or satellite images created using proprietary data, such as Google software (Google Maps, Street View, and Earth). For more information, see our copyright guidelines: http://journals.plos.org/plosone/s/licenses-and-copyright.

Response: We acknowledge the editor suggestion and have removed Fig 1 from the revised manuscript . 

Reviewers' comments: 

Reviewer #1: 

1. The manuscript focus on human criptosporidiosis in Ethiopia by comparing different standard and molecular techniques. Results are impressive and deserves of being published, given their importance.

My only suggestions is to improve comparision with other realities and also in one health approach, considering also animal reservoir. These recent papers can be included into the discussion.

Line 258 and after:

Dessì, G., Tamponi, C., Varcasia, A., Sanna, G., Pipia, A.P., Carta, S., Salis, F., Díaz, P., Scala, A. Cryptosporidium infections in sheep farms from Italy(2020) Parasitology Research, 119 (12), pp. 4211-4218.

Díaz, P., Varcasia, A., Pipia, A.P., Tamponi, C., Sanna, G., Prieto, A., Ruiu, A., Spissu, P., Díez-Baños, P., Morrondo, P., Scala, A.Molecular characterisation and risk factor analysis of Cryptosporidium spp. in calves from Italy(2018) Parasitology Research, 117 (10), pp. 3081-3090.

Response: Thank you for the suggestion. We have expanded the discussion that compares our findings with others and added the suggested references. We have also commented on the importance of one health approach to control zoonotic cryptosprordiasis (line 266-275). The revised text reads as follows ‘“Altogether, these findings may suggest human and animals as reservoirs for the C. parvum [49,50]. Thus, an integrated, transdisciplinary and multilevel one health approach strategies/intervention that target humans, animals and their shared environment/transmission routes would be necessarily to effectively control cryptosporidiosis in regions endemic for C. parvum infection [51]. An integerated research involving veterinary, public health and environmental fields would help better understand the burden, risk factors, transmission routes of the zoonotic C. parvum infection and plan collaborative one health approach to treat or prevent infection in animals and humans, reduce environmental contamaination and block transmission in endemic regions [51]”. 

Reviewer #2: 

Study provides data on the prevalence and genetic diversity of Cryptosporidium in human patients in Ethiopia. Study surely fills a gap I knowledge on Cryptosporidium species involved in human cryptosporidioses in the area, however, relatively small sample set does not allow for robust interpretation. Considering this fact, some parts of the study seem a bit over-interpreted, mainly when discussing zoonotic vs. anthroponotic transmission. It is questionable, how much the fact, that some of the genotypes are shared among animals and humans, suggests an infection from animal reservoir. For such an assumptions, results from animals living in proximity of tested subjects can be of great value.

Relatively high number of samples proved positive only in NAATs, which opens a question of passive transfer of the oocysts through human digestive system. For humans living in highly contaminated environment this can be expected. This part might deserve some explanation or justification… otherwise the “true” prevalence mentioned might be the “distorted one” (Line 198).

Response: We acknowledge your suggestions and have revised the discussion to avoid over interpretation of the current finding related to zoonotic/anthropmetric transmission.We have also discussed the reasons for the low prevalence of infection detected through the microscope (line 238-251).

2. Paragraph in lines 308-315 raises several doubts about part of the results and mainly about conclusiveness of the study. In my opinion, its placement before Conclusions without explanations is misleading. This deserves some attention and explanations… possibly in respective parts of the results and discussions.

Response: We thank the reviwer for this suggestion. We have rephrased the statement in the paragraph from the revised manuscript in line # 297- 300

3.Line 55: consider replacing of “for identifying” by “for detection”

 Response : we have replaced ‘ identifying’” by ‘detecting’ in line 54 

4. Line 66-67: mentioning fairly distant and non-zoonotic abomasal C. andersoni without any explanation is confusing.

Response: We have revised the text in line 66-67. It read as “Indeed, a study reported Cryptosporidium parvum among 35 (87.5%) pre-weaned calves specimens examined in central Ethiopia” 

5. Line 71: replace “genetic distribution” by more proper term

Response : We have replaced the phrase ‘genetic distribution” with “ genetic diversity “ ( line # 71 ) 

6. Line 75: the goal “identifying genetic diversity… … in Ethiopia” is over-ambitious, considering the small number of samples from a single geographic region…

Response: We have revised the objectives of the study in the background considering the nature of the samples( line # 76). The revised text reads as ‘identify genetic diversity of the parasite circulating in human population living in Wurgissa district and Hawassa in Ethiopia”.

7. Line 99 (and some other places): double check using the upper case letters

Response: we have doubled check all the words in the manuscript for proper capitalization 8. “55.1% reported gastrointestinal symptoms with diarrhea”… not clear to me, if this was in time of sample collection or in “any” time before visiting the hospital 

Response; the patients reported they have had diarrhea perior to their health center visit [ line # 170-171). The revised text reads as ‘Of the adults, 55.1% reported gastrointestinal symptoms with diarrhea perior to visiting the health center.”

9. Line 179:”infection” not Italic

Response : we have removed the italization for the word infection in line 173.. 

10. Line 241-245: it is always high to compare data from various studies with various methodology and robustness of diagnostics, however, the option, that lower prevalences can be attributable to lower performance of diagnostic procedures deserves mentioning, before searching for other explanations.

Response: We have revised the text that explain the reason for the higher prevalence of infection in the current study compared to the reports in the other regions of the Ethiopia. The revised text reads as (line # 237-239“. The lower prevalence of cryptosporidiosis reported in the aforementioned studies in Ethiopia could be due to the less sensitivity of the microscopy procedures used for diagnosing Cryptospordium infection [26]´. 

11. Line 249-252: not clear to me, why zoonotic transmission is considered more plausible than human-to-human

Response: Thank you picking this. We have commented on the risk of human to human transmission which might have contribuited to the increased prevalence of infection in the current study area. The revised text reads as “ Altogether, these findings may suggest human and animals as reservoirs for the C. parvum [49,50]. Thus, an integrated, transdisciplinary and multilevel one health approach strategies/intervention that target humans, animals and their shared environment/transmission routes would be necessarily to effectively control cryptosporidiosis in regions endemic for C. parvum infection [51].” (line 266-270)

12. Line 252-253 requires references

Response: we have included a reference [ line #264-265]. The revised text reads as ‘in human samples from Egypt, Ethiopia, and Malaysia [15, 44-46] ”

13. Line 270-272: you say “Moreover, this subtype was recorded in immunocompetent adults in Slovakia [42] and children in Mexico [43], which provides evidence for its role in animal-to-human transmission”… this is not clear… how this relates to animal-to-human transmission? Please, clarify…

Response. We acknowledge the reviewers comment and have rewritten in the text (line 259-261) “ The IIaA16G1R1 subtype was reported in lamb, calves, and humans, as well as water sources in Romania [36], Estonia [37], and Slovakia [38].”. 

14. Lines 272-274: the statement about sylvatic transmission is purely speculative in given context, mentioning feral horses but also (probably?) domestic calves and goats.

Response. We have revised the text to read as “….Altogether, these findings may suggest human and animals as reservoirs for the C. parvum [49,50]. Thus, an integrated, transdisciplinary and multilevel one health approach strategies/intervention that target humans, animals and their shared environment/transmission routes would be necessarily ….

15. Lines 283-302: this deserves some shortening, considering the sample set size and low significance of the results. In discussed aspects the study revealed nothing new.

Response: We thank the reviwer for this suggestion. We have shortned the text on the discussion including lines 283-302 the shortened paragraph read as “…. In addition, as genotyping was performed for small human samples, confirmation of zoonotic transmission was limited in this study….” 

16. Last but not least, please, pay attention to formatting of references

Response: we have formatted the references following the Plos one authors guideline

---

## [Decision Letter · Decision Letter 1]

31 May 2021

Molecular characterization of Cryptosporidium spp. from humans in Ethiopia

PONE-D-21-06475R1

Dear Dr. Woreta,

We’re pleased to inform you that your manuscript has been judged scientifically suitable for publication and will be formally accepted for publication once it meets all outstanding technical requirements.

Kind regards,

Maria Stefania Latrofa

Academic Editor

PLOS ONE

Reviewers' comments:

Reviewer's Responses to Questions

**Comments to the Author**

1. If the authors have adequately addressed your comments raised in a previous round of review and you feel that this manuscript is now acceptable for publication, you may indicate that here to bypass the “Comments to the Author” section, enter your conflict of interest statement in the “Confidential to Editor” section, and submit your "Accept" recommendation.

Reviewer #3: All comments have been addressed

2. Is the manuscript technically sound, and do the data support the conclusions?

Reviewer #3: Yes

3. Has the statistical analysis been performed appropriately and rigorously? 

Reviewer #3: Yes

4. Have the authors made all data underlying the findings in their manuscript fully available?

Reviewer #3: Yes

5. Is the manuscript presented in an intelligible fashion and written in standard English?

Reviewer #3: Yes

6. Review Comments to the Author

Reviewer #3: Dear author

The manuscript is impressive and deserve of being published, given their importance. My only suggestion is to refer to an animal reservoir. If you had been worked on animal type besides Human type maybe gave more information and clear at the site of study. The manuscript has robust interpretation and unambiguous for Cryptosporidium species distribution that infecting Human beings in both sites.

7. PLOS authors have the option to publish the peer review history of their article (what does this mean?). If published, this will include your full peer review and any attached files.

Reviewer #3: No

---

## [Editor Report · Acceptance letter]

4 Jun 2021

PONE-D-21-06475R1 

Molecular characterization of *Cryptosporidium* spp. from humans in Ethiopia. 

Dear Dr. Hailu:

I'm pleased to inform you that your manuscript has been deemed suitable for publication in PLOS ONE. Congratulations! Your manuscript is now with our production department. 

Kind regards, 

on behalf of

Dr. Maria Stefania Latrofa 

Academic Editor

PLOS ONE